# Metabolic Health Together with a Lipid Genetic Risk Score Predicts Survival of Small Cell Lung Cancer Patients

**DOI:** 10.3390/cancers13051112

**Published:** 2021-03-05

**Authors:** Lara P. Fernández, María Merino, Gonzalo Colmenarejo, Juan Moreno Rubio, Tais González Pessolani, Guillermo Reglero, Enrique Casado, María Sereno, Ana Ramírez de Molina

**Affiliations:** 1Molecular Oncology Group, IMDEA Food Institute, CEI UAM + CSIC, E-28049 Madrid, Spain; jmrubio@salud.madrid.org (J.M.R.); guillermo.reglero@imdea.org (G.R.); 2Medical Oncology Department, Infanta Sofía University Hospital, San Sebastián de los Reyes, E-28709 Madrid, Spain; mmerinos.hulp@salud.madrid.org (M.M.); enrique.casado@salud.madrid.org (E.C.); maria.sereno@salud.madrid.org (M.S.); 3Biostatistics and Bioinformatics Unit, IMDEA-Food Institute, CEI UAM + CSIC, E-28049 Madrid, Spain; gonzalo.colmenarejo@imdea.org; 4Pathology Department, Infanta Sofía University Hospital, San Sebastian de los Reyes, E-28709 Madrid, Spain; tais.gonzalez@salud.madrid.org

**Keywords:** small cell lung cancer, prognosis, lipid metabolism, gene expression profile, metabolic health, high stage tumors

## Abstract

**Simple Summary:**

Despite the progress in surgery and therapies, small cell lung cancer (SCLC) is still one of the most lethal types of cancer. The disease control remains heterogeneous and consequently, the ability to predict patient survival would be of great clinical value. Here, we propose for the first time, a metabolic precision approach for SCLC patients. We found that a healthy metabolic status contributes to increasing SCLC survival. Moreover, we discovered that two lipid metabolism-related genes, racemase and perilipin 1, and a genetic risk score of both genes, predict better SCLC survival. Our results show that a metabolic scenario characterized by metabolic health, lipid gene expression and environmental factors, is crucial for increase SCLC survival.

**Abstract:**

Small cell lung cancer (SCLC) prognosis is the poorest of all types of lung cancer. Its clinical management remains heterogeneous and therefore, the capability to predict survival would be of great clinical value. Metabolic health (MH) status and lipid metabolism are two relevant factors in cancer prevention and prognosis. Nevertheless, their contributions in SCLC outcome have not yet been analyzed. We analyzed MH status and a transcriptomic panel of lipid metabolism genes in SCLC patients, and we developed a predictive genetic risk score (GRS). MH and two lipid metabolism genes, racemase and perilipin 1, are biomarkers of SCLC survival (HR = 1.99 (CI95%: 1.11–3.61) *p* = 0.02, HR = 0.36 (CI95%: 0.19–0.67), *p* = 0.03 and HR = 0.21 (CI95%: 0.09–0.47), respectively). Importantly, a lipid GRS of these genes predict better survival (c-index = 0.691). Finally, in a Cox multivariate regression model, MH, lipid GRS and smoking history are the main predictors of SCLC survival (c-index = 0.702). Our results indicate that the control of MH, lipid gene expression and environmental factors associated with lifestyle is crucial for increased SCLC survival. Here, we propose for the first time, a metabolic precision approach for SCLC patients.

## 1. Introduction

Despite the progress of surgery and therapies, lung cancer is still one of the most lethal types of cancer according to the World Health Organization (WHO) [1]. Small cell lung cancer (SCLC), a lung cancer subtype of neuroendocrine origin, accounts for 15–20% of lung tumors. SCLC exhibits rapidly metastatic spread and patient prognosis in this subtype is the poorest of all types of lung cancer [2]. The disease control remains heterogeneous and consequently, the ability to predict patient survival would be of great clinical value.

The alteration of cellular metabolism is one of the hallmarks of cancer. Cancer cells modify their metabolic properties to provide additional energy and anabolic demands in order to maintain cell proliferation and dissemination [3]. In addition to well-known Warburg effect, lipid metabolism contributes to cancer metabolic reprogramming [4]. Interestingly, lipid metabolism-related genes display remarkable roles as prognostic biomarkers in various types of cancer including non-small cell lung cancer [5,6,7,8]. However, and probably due to its low incidence and particular characteristics, the use of lipid metabolism-related genes to predict patient prognosis in SCLC remains unexplored.

The metabolic status of cancer patients clearly determines their clinical outcome. Compelling evidence sustains that systemic as well as local metabolic conditions have a considerable effect on tumor development, as well as on body composition and organ functions [9]. Therefore, metabolic health (MH)—a multifactorial condition that includes parameters like obesity, dietary patterns, gene–diet interactions and lipid metabolism—arises as a relevant factor in cancer prevention and prognosis [10]. Importantly, the contributions to MH and lipid metabolism on SCLC outcome have not yet analyzed.

In this study, a panel of lipid metabolism-related genes and their relationship with survival and patient’s metabolic status was successfully used as marker for SCLC prognosis and future therapeutic monitoring. We propose, for the first time, a metabolic precision approach for high-grade SCLC patients.

## 2. Materials and Methods

### 2.1. Patient’s Selection and Gene Expression Analysis

A total number of 61 formalin-fixed, paraffin-embedded (FFPE) samples were acquired from SCLC patients of the Medical Oncology Service of Infanta Sofía University Hospital (San Sebastián de los Reyes, Madrid, Spain). SCLC patients enrolled from 1 November 2008 to 31 September 2017, were included in this analysis. Clinical, physiological and pathological data were collected from medical reports (Figure 1A). This study was approved by the IMDEA Food Research Ethics Committee (Ethical code: IMD: PI-0013). Overall survival (OS) was defined from the date of diagnosis to the date of patient death.

Gene selection and the analysis of gene expression was performed as previously described [5,7]. We used RNeasy FFPE Kit (Qiagen, Germantown, MD, USA) to obtain total RNA from samples previously deparaffinated, 1 μg of RNA was reverse transcribed by High Capacity cDNA Archive Kit (Applied Biosystems, Waltham, MA, USA) for 2 h at 37 °C. A Taq-Man Low Density Array (Applied Biosystems) was designed for this experiment and was composed of 22 lipid metabolism-related genes, selected due to their role as regulators of cell metabolism [5,7]. The transcriptomic panel was preliminarily tested in 37 SCLC patients. Then, the significant associations were analyzed in the complete dataset of patients (*n* = 61). Gene-expression assays were performed in a HT–7900 Fast Real time PCR.

### 2.2. Statistical Analysis

Statistical analysis was executed as previously described [5,7]. Gene expression was quantified with the 2–ΔCt method. The Kaplan–Meier method was used to estimate overall survival (OS) curves, and univariate Cox regression analyses were used to test the association between OS and clinical variables. Multivariable Cox regression models were derived to estimate the hazard ratio (HR) of gene expression and metabolic health, and adjusting for potential confounding factors. Each gene expression variable was binarized based on the maximally selected rank statistics and using Log-rank scores [11]. A genetic score was derived as the sum of genes with high-binned expression. The Bonferroni method was applied for multiple test correction. Conditionally adjusted survival curves were generated with the *survminer* package. Statistical significance was defined as the *p* value <0.05. The models were validated through bootstrap resampling (500 resamples) and using the optimism-corrected c-index as a measure of predictive capacity [12]. The latter is an extension of the AU-ROC (area under the receiving operating characteristic curve) in binary outcomes to censored data. All the analyses were performed using the R statistical software (version 3.6.1, R Foundation for Statistical Computing, Vienna, Austria) (www.r-project.org (accessed on 1-12-2020)).

## 3. Results

### 3.1. A Healthy Metabolic Status Contributes to Increase SCLC Survival

We explored the contribution of MH to SCLC outcome. The clinical characteristics of all patients are described in Figure 1A. The estimated HRs for the OS of clinical characteristics based on univariate Cox models are also shown. As expected, the Eastern Cooperative Oncology Group (ECOG) performance status and metastasis were associated with a higher risk of death with HRs of 1.77 (CI95%: 1.2–2.61) and 1.82 (CI95%: 0.984–3.38), respectively. Interestingly, the presence of diabetes mellitus and metformin treatment also were associated with an increased risk of decease (HRs of 3.71 (CI95%: 1.68–8.19) and 3.56 (CI95%: 1.52–8.36) respectively). By contrast, body mass index (BMI) showed a trend to be a favorable marker of survival with a HR of 0.97 (CI95%: 0.926–1.01). Unfortunately, we were not able to detect any association with cholesterol levels, probably due to the low number of patients for which these data were recorded (Figure 1A).

We classified our SCLC patients according to their MH status. Metabolically healthy (MH) patients (*n* = 30) were essentially those without metabolic diseases (without cardiovascular diseases or statins treatment, with HDL levels equal or higher than 40 mg/dL in men or 50 mg/dL in women [13] and without diabetes (DM) or metformin treatment). By contrast, metabolically unhealthy (MU) patients (*n* = 31) were those with cardiovascular diseases or statins treatment, with HDL levels lower than 40 mg/dL in men or 50 mg/dL in women or those with diabetes or following metformin treatment (Figure 1B). MH is an independent marker of SCLC survival (HR = 1.99 (CI95%: 1.11–3.61) *p* = 0.02, c-index = 0.604) (Figure 1C) and metabolically healthy (MH) patients showed higher SCLC OS than metabolically unhealthy (MU) patients (Figure 1D).

### 3.2. Racemase and Perilipin Expression Levels Are Prognostic Markers of Increased Survival in SCLC Patients

The deregulated gene expression of certain lipid metabolism genes has been associated with altered cell metabolism and poor survival in several types of cancer and as a result they have been proposed as prognostic biomarkers [5,6,7,8,14]. We investigated the prognostic value of lipid metabolism genes in SCLC patients, by performing survival analysis based on gene expression.

A transcriptomic panel of 22 lipid metabolism genes [7] was preliminarily tested in 37 SCLC patients and after adjusting the analysis for putative confounder variables (age, sex, stage and smoking history) we found statistically significant associations after correction for multiple tests with OS for two genes: Alpha-Methylacyl-CoA Racemase (*AMACR*) and Perilipin 1 (*PLIN1*) (Figure 2A). Then, we analyzed AMACR and PLIN1 expression in the complete dataset of patients (*n* = 61). Kaplan–Meier plots for the OS of both genes showed a significant association between high gene expression and better clinical outcome (Figure 2B).

*AMACR* encodes an enzyme, localized in both mitochondria and peroxisomes, that interconverts pristanoyl-CoA and C27-bile acylCoAs between their (R)- and (S)-stereoisomers. The (S)-stereoisomers are necessary for peroxisomal beta-oxidation [15]. *PLIN1* encodes for a protein that coats lipid storage droplets protecting them from the action of lipase. PLIN1 is the major cAMP-dependent protein kinase substrate in adipocytes and, when it is unphosphorylated, blocks lipolysis [16] (Figure 2C). Thus, both enzymes are involved in a proper metabolic function of the cells.

A lipid genetic risk score (GRS) based on the expression of these two lipid metabolism-related genes was developed, as a sum of high-binned expressions. In a Cox multivariable regression model with the full sample, adjusted hazard ratios for individual AMACR and PLIN1 gene expression were 0.36 (CI95%: 0.19–0.67), *p* = 0.03 and 0.21 (CI95%: 0.1–0.47), *p* = 0.009 with a bootstrap C-index of 0.650 and 0.664, respectively. Importantly, the lipid GRS of these two genes was able to significantly predict better the survival of SCLC (adjusted HR was 0.388 (CI95%: 0.249–0.603) *p* = 0.0008, with an improved c-index of 0.691) (Figure 2D). Conditionally adjusted survival curves for the three values of GRS are shown in Figure 2E.

All these data consistently corroborate our findings that the upregulation of racemase and perilipin is associated with better survival in SCLC patients. Importantly, those SCLC patients that express high levels of racemase and perilipin increased their survival time to almost 7 months.

### 3.3. Metabolic Health, Lipid Metabolism Genetic Risk Score and Smoking Behavior Are the Main Predictors of SCLC Survival

Given the role of MH and AMACR and PLIN1 expression on SCLC survival, we decided to generate a predictive model considering all potential confounder factors. In a Cox multivariate regression model, MH, lipid GRS and smoking history were independent risk factors for SCLC outcome, with the most predictive bootstrap c-index of 0.7023**** (Figure 2F). Our results prove that a metabolic scenario characterized by MH, lipid gene expression and environmental factors associated with lifestyle such as smoking behavior, is crucial for increasing SCLC survival.

## 4. Discussion

SCLC represents a rare and particularly aggressive form of lung cancer. Nowadays precision nutrition and precision medicine open new avenues to fight against cancer and its clinical management. Here, we analyzed for the first time a panel of metabolic genes together with patient metabolic status and their relationship with SCLC patient’s survival. We were able to detect a robust metabolic predictive marker for patient outcome.

It is extensively known that once that tumor is diagnosed, the metabolic status of the patient plays a key role in prognosis [9]. In this study, we described for the first time that MH is an independent prognostic factor for SCLC outcome. Due to the relevant link between MH and tumor progression, the effective control of the metabolic status of individuals might represent a specific approach to follow cancer progression. Moreover, in several types of cancer, obesity and consequently, BMI have a negative effect on patient´s outcome. However, in certain tumor types, some of them of neural origin, a good metabolic performance, including high BMI, could be beneficial. In our study, BMI showed a trend to be a favorable marker of survival. This fact is known as the paradox of obesity [17].

Several enzymes related to lipid-metabolic pathways have been associated with cancer survival and proposed as biomarkers of cancer [8]. Here, we described a lipid GRS based on the expression of two lipid metabolism-related genes involved in the metabolic function of the cell: *AMACR* and *PLIN1*. Although the findings were highly internally consistent, the sample size of this study was relatively limited. However, the bootstrap validation, where the model was repeatedly derived in the samples with replacement of the original sample, and then evaluated in the latter, show a good predictive performance on future subjects from the resulting optimism-corrected c-indexes, which also increased upon the inclusion of the MH variable, indicating no overfit.

An association of AMACR protein expression with better SCLC prognosis has been reported previously, adding value to our findings [15]. Nevertheless, this is the first time that *AMACR* mRNA expression has been described as a SCLC biomarker. *PLIN1* expression has also been reported as a favorable marker of other types of cancer like breast cancer but not for SCLC [18]. The design, validation and use of genetic scores unlock new opportunities to incorporate precision approaches into clinical advice [8]. Notably, patients overexpressing *AMACR* and *PLIN1* increased their survival time to almost 7 months. Smoking behavior could interfere with lipid metabolism. Importantly, we developed a predictive model that encompasses three independent risk factors for SCLC outcome: MH, lipid GRS and smoking history.

All these data allow us to propose that, in SCLC, metabolic health, *AMACR* and *PLIN1* expression and smoker status are defining a specific metabolic scenario, associated with better patient survival. Our goal is to identify a predictive metabolic model of survival, providing novel precision strategies for SCLC patients, including targeted nutrition to regulate MH. The use of this information will allow a better stratification of patients which will be translated into improvements on the treatment and overall survival of the patients.

## 5. Conclusions

In this work, we identified a predictive metabolic model of SCLC prognosis that will be of great clinical value. We found that the metabolic status of SCLC patients clearly determines their clinical outcome. In addition, we determined that two lipid metabolism genes, racemase and perilipin 1, and a genetic risk score of both, also modulate SCLC survival. Here, we proposed for the first time, a metabolic precision approach for SCLC patients based on the control of metabolic health, the lipid-metabolism gene expression and lifestyle.

## Figures and Tables

**Figure 1 cancers-13-01112-f001:**
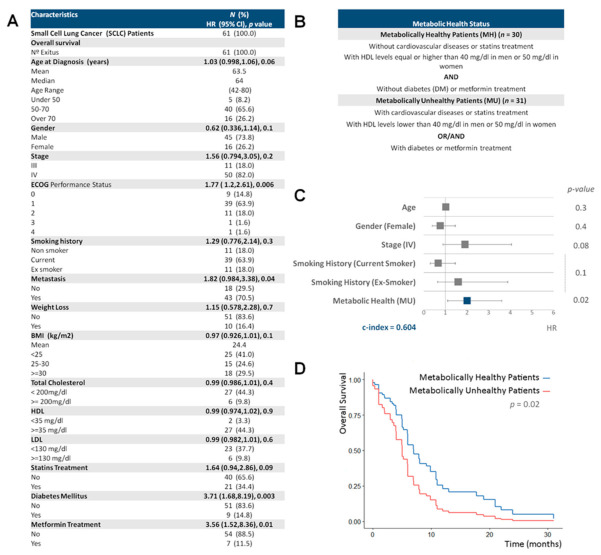
Associations between small cell lung cancer (SCLC) survival and metabolic health. (**A**) Clinical characteristics of 61 small cell lung cancer patients. Univariate Cox regression analysis was used to test the association between overall survival and clinical variables. BMI: body mass index, ECOG: eastern cooperative oncology group, HDL: high-density lipoprotein and LDL: low-density lipoprotein. (**B**) Participant characteristics by metabolic health status. (**C**) Forest plot showing the hazard ratio (HR) and 95% confidence interval (CI) estimates for the overall survival (OS) of clinical characteristics and metabolic health status in 61 SCLC patients. HR, their 95%CI and *p*-values were calculated from multivariable Cox regression model; the latter were obtained after a marginal likelihood-ratio test for each factor, further corrected for multiple test by using the Bonferroni method. (**D**) Conditionally adjusted survival curves for SCLC overall survival by metabolic health status from the previous model.

**Figure 2 cancers-13-01112-f002:**
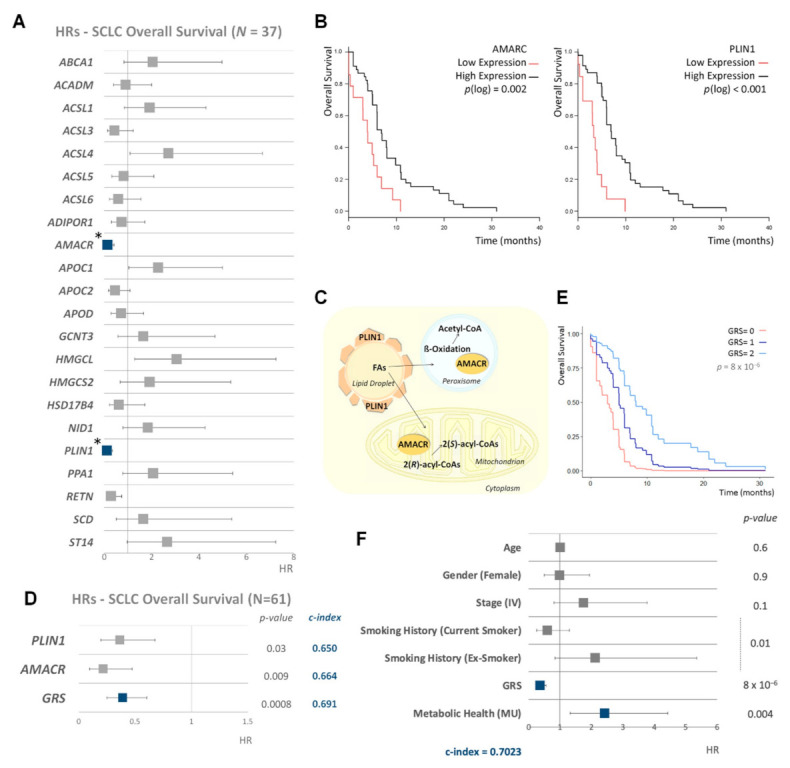
Impact of lipid metabolism and metabolic health on the survival of small cell lung cancer (SCLC) patients. (**A**) Forest plot showing the hazard ratio (HR) and 95% confidence interval (CI) estimates for the overall survival (OS) of lipid metabolism binarized gene expression in 37 SCLC patients from our dataset. Multivariable Cox regression model was used to estimate the hazard ratio of gene expression, and adjusting for potential confounding factors: age, sex, stage and smoking status. *p*.values were obtained from marginal likelihood-ratio tests of each factor and corrected by Bonferroni method. (**B**) Kaplan–Meier plots for racemase (*AMACR*) and perilipin 1 (*PLIN1*) expression in 61 SCLC patients. Each gene expression variable was binarized by determining a cutpoint based on maximally selected rank statistics and using Log-rank scores. p(log): *p*-value Log-rank from log-rank test. (**C**) Schematic representation of intracellular functions of AMACR and PLIN1 enzymes. (**D**) Forest plot showing the hazard ratio (HR) and 95% confidence interval (CI) estimates for overall survival (OS) of *AMACR* and *PLIN1* expression and genetic risk score (GRS) for both genes in SCLC. Multivariable Cox regression model was derived to estimate the hazard ratio of gene expression and adjusting for potential confounding factors. The Bonferroni method was applied for the multiple test correction of the *p*-values. (**E**) Conditionally adjusted survival curves for SCLC overall survival by lipid GRS. (**F**) Forest plot showing the hazard ratio (HR) and 95% confidence interval (CI) estimates for the overall survival (OS) of clinical characteristics, lipid GRS and metabolic health status in 61 SCLC patients. HR, their 95%CI and *p*-values were calculated from multivariable Cox regression model. *p*.values were obtained from marginal likelihood-ratio tests of each factor and corrected by Bonferroni method. MU: metabolically unhealthy patients. * Adjusted and corrected *p*-value < 0.05.

## Data Availability

The data presented in this study are available on request from the corresponding author. The data are not publicly available due to ethical restrictions.

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
