# Peer review of "Metabolic Health Together with a Lipid Genetic Risk Score Predicts Survival of Small Cell Lung Cancer Patients"

_cancers, 2021, doi:10.3390/cancers13051112_

Round 1
Reviewer 1 Report
The paper presents very intersting findings, but some information and improvments should be done.
- There is no information for how long time the samples was collected in the Hospital (it could explain the number of cases).
- In maerial and methods there is 67 number of cases, but in section 3.2 there are only 37 cases. Please clarify.
- In all figure panels where is the survival curves there should be p value added to present the results more clearly.
- There is no control group. Has it been analyzed?
Author Response
Thank you very much for your interest and approval. We really appreciate your valuable suggestions. We have revised our manuscript accordingly. Please find attached revised version of the manuscript, which includes text modifications. All changes are highlighted in yellow.
1. There is no information for how long time the samples were collected in the Hospital (it could explain the number of cases).
SCLC patients enrolled from through November 1, 2008, to September 31, 2017 were included in this analysis. We have added this information in the Materials and Methods section in order to clarify this point (Line 67).
2. In material and methods there is 67 number of cases, but in section 3.2 there are only 37 cases. Please clarify.
In Results section we described that the transcriptomic panel of 22 lipid metabolism genes was preliminarily tested in 37 SCLC patients (line 137). Then, the significant associations were analyzed in the complete dataset of patients (line 141). In order to clarify that, we have also included this information in the Materials and Methods section (line 89).
3. In all figure panels where is the survival curves there should be p value added to present the results more clearly.
We agree with the comment. We have added p-values in all survival curves.
4. There is no control group. Has it been analyzed?
There is no control group without disease. The patients were divided into high- and low- gene expression and we compared overall survival between these two groups of patients. Each gene expression variable was binarized based on maximally selected rank statistics and using Log-rank scores in two groups, high and low gene expression. We specified this in Statistical Analysis section.
Reviewer 2 Report
The paper submitted by Dott. L. Fernandez deals with the alterations of lipid metabolism in small cell lung cancer patients.
In this study a correlation between some genes involved in lipid metabolism, whose changes could be considered as potential risk factor or/and could favor OS, and metabolic health has been analyzed.
The paper is well organized and it sounds good.
However, I have just few perplexities.
Transcriptomic analysis obviously could allow the identification of more than two genes, as it concerns this paper it allowed the identification of AMACR and PLIN1, it is not clear to me why the Authors choose to evaluate only these two genes.
The authors should spend few worlds to explain why in Table 1, when they present the clinical values of Total cholesterol, HDL and of LDL the number of patients is not 61?
In addition, remaining on Table1, section C: why the authors specified the gender (female), we should image that there is a hormonal influence?, but female are only the 26.2 percent of analyzed samples.
Smoke could interfere with lipid metabolism, does a history of smoking (either current or ex) have any effect on AMAC racemase or Perilipin activity/expression, the authors should comment this point.
Author Response
Thank you for your comments, we appreciate your valuable suggestions. We have revised our manuscript accordingly. Please find attached revised version of the manuscript, which includes text modifications
- Transcriptomic analysis obviously could allow the identification of more than two genes, as it concerns this paper it allowed the identification of AMACR and PLIN1, it is not clear to me why the Authors choose to evaluate only these two genes.
We performed transcriptomic analysis of 22 lipid metabolism-related genes, selected due to their role as regulators of cell metabolism (Vargas, T. et al. ColoLipidGene: Signature of Lipid Metabolism-Related Genes to Predict Prognosis in Stage-II Colon Cancer Patients. Oncotarget 2015, 6, 7348–7363 and Fernández, L.P.; et al. Metabolic Enzyme ACSL3 Is a Prognostic Biomarker and Correlates with Anticancer Effectiveness of Statins in Non-Small Cell Lung Cancer. Mol. Oncol. 2020, doi:10.1002/1878-0261.12816.).
Of these 22 genes, we found statistically significant associations with OS after correction for multiple tests in only two genes: Alpha-Methylacyl-CoA Racemase (AMACR) and Perilipin 1 (PLIN1).
- The authors should spend few worlds to explain why in Table 1, when they present the clinical values of Total cholesterol, HDL and of LDL the number of patients is not 61?
Clinical, physiological and pathological data were collected from medical reports. In the case of cholesterol, HDL and LDL, unfortunately, this data was only available in 33, 29 and 29 patients respectively. We have added a comment in this respect in Results section (Line 117).
- In addition, remaining on Table1, section C: why the authors specified the gender (female), we should image that there is a hormonal influence?, but female are only the 26.2 percent of analyzed samples.
Sex is considered as a potential confounding factor (among other reasons for hormonal influences, as you mentioned). However, we did not detect an effect of sex (female) in SCLC survival (p-value= 0.4).
As potential confounding factors of risk, it is mandatory to include at least age and sex. In SCLC, stage and smoker history are also relevant as confounding factors. That is the reason why we have included these clinical characteristics together with metabolic health status in our analysis.
- Smoke could interfere with lipid metabolism, does a history of smoking (either current or ex) have any effect on AMAC racemase or Perilipin activity/expression, the authors should comment this point.
Our results shown that AMACR and PLIN1 expression are associated with SCLC prognosis independently of confounder variables (age, sex, stage and smoking history) (Fig2. Panel A).
Then when we developed a model of SCLC risk in which we include all putative relevant variables, we also found that AMACR and PLIN1 expression (measured by Genetic Risk Score-GRS-), metabolic health and smoking behavior, are independent and crucial factors for SCLC survival (Line 172) (Fig2. Panel F).
Following your suggestion, we have included a paragraph in the discussion section commenting this (Line 221).
Round 2
Reviewer 1 Report
Paper could be accepted for printing. Thank you for correcting the work in accordance with the comments of the previous review.